

# Decreasing foraminiferal flux in response to ongoing climate change in the Santa Barbara Basin, California

Emily Havard[1], Katherine Cherry[1], Claudia Benitez-Nelson[2], Eric Tappa[2], Catherine V. Davis[1]

[1]Department of Marine, Earth, and Atmospheric Sciences, North Carolina State University, Raleigh, NC 27606, USA

[2]School of the Earth, Ocean, and Environment, University of South Carolina, Columbia, SC 29208, USA

*Correspondence to*: Emily Havard (ehavard@ncsu.edu)

**Abstract.** The rapid response of foraminiferal assemblages to changing climate makes their shells an invaluable geological record of the past. However, the time frame over which foraminifera respond to climatic signals and the specific drivers influencing assemblage composition and abundance remain obscure. We focus on the impact of ongoing, anthropogenic

climate change on planktic foraminifera in the California Current ecosystem, which would appear as a nearly instantaneous event in the sediment record. The Santa Barbara Basin sediment trap, located off the coast of California, USA since 1993, provides a 28-year record of particulate and foraminiferal flux to the basin's seafloor. The sediment trap captures the superposition of the annual cycle of seasonal upwelling, Pacific multiannual El Niño-Southern Oscillation-driven temperature changes, and anthropogenically forced climate change. We present data on planktic foraminiferal flux collected between 2014-

2021, at two-week intervals (164 samples, 60,006 individuals) and compare results to previously published data from 1993-1998. Consistent with previous studies, the most abundant species from 2014-2021 were *Globigerina bulloides*, *Neogloboquadrina incompta*, and *Turborotalita quinqueloba*, with peak fluxes occurring in the spring and summer. Lower fluxes and an increase in the abundance of *N. incompta* and subtropical species characterize the winter season. We find a 37.9 % decrease in total foraminiferal flux relative to the 1990s, primarily driven by a decrease in *G. bulloides* abundance. This

decrease is accompanied by a 21.9 % overall reduction in calcium carbonate flux. We also find a decrease in the relative abundance of subtropical species (*Globigerinoides ruber*, *Orbulina universa*, *Neogloboquadrina dutertrei*) and their fluxes compared to the 1990s, contrary to expectations if assemblages and fluxes were to follow anthropogenic warming signals. We hypothesize that the observed decrease in subtropical species abundance and flux is likely related to an increase in acidification and in the timing and magnitude of upwelling along the California coast. The extremely rapid responses of foraminifera to

ongoing changes in carbonate chemistry and temperature suggest that climate change is already having a meaningful impact on coastal carbon cycling. The observed decrease in particulate inorganic carbon (PIC) flux relative to particulate organic carbon (POC) flux may facilitate increased oceanic uptake of atmospheric $CO_2$.





## 1 Introduction

Over the last three decades, the annual mean atmospheric $CO_2$ concentration has increased by ~ 64 ppm, driving higher sea
surface temperatures (SST) and acidifying the ocean, negatively impacting marine ecosystems (Keeling et al., 2001; Nagelkerken and Connell, 2015; Lan and Keeling, 2024). Foraminifera are protists that form a calcium carbonate shell and are frequently preserved in marine sediments (Hendy et al., 2004; Field et al., 2006; Kucera, 2007; White et al., 2013; Schiebel and Hemleben, 2017). On geological time (> 1000 years) and global spatial scales, foraminiferal communities respond rapidly to climate change (Kucera et al., 2005; Morey et al., 2005; Field et al., 2006; Jonkers et al., 2019). As a result they have been
used to study oceanographic and climatic change throughout the Phanerozoic and have greatly contributed to the understanding of major climatic events through analysis of their shell chemistry and species distributions (Lisiecki and Raymo, 2005; Kucera, 2007; Schiebel and Hemleben, 2017; Jonkers et al., 2019) including climate change associated with rapid carbon release and warming such as the Paleocene-Eocene Thermal Maximum (~56 Mya) (Thomas and Shackleton, 1996; Zachos et al., 2003; McInerney and Wing, 2011). Moreover, marine calcifiers such as foraminifera and coccolithophorids are an important part of
the marine carbon cycle and comprise 20-80% of calcite flux to the deep ocean (Schiebel, 2002; Schiebel et al., 2007). Understanding how foraminifera respond to geologically rapid environmental change therefore has important implications for the future carbon cycle, while also providing context for reconstructions of past climate that rely on geochemical and assemblage records of foraminifera.

Foraminiferal assemblages have already shifted in response to local temperature change, such that an assemblage found in the tropics during pre-industrial times now exists at higher latitudes in the late 20th century (Jonkers et al., 2019), a trend also observed in other types of organisms such as tropical corals (Yamano et al., 2011) and barnacles (Crickenberger and Wethey, 2018). Foraminiferal assemblage records from Santa Barbara Basin (SBB) sediments exhibit an increase in the abundance of tropical and subtropical species and a decrease in temperate and subpolar species throughout the warming 20th century (Field
et al., 2006). However, temperature is not the sole driver of ecological changes. Rather, foraminifera are impacted by a wide range of factors that vary on multiple timescales. Changes in pH, stratification, temperature, upwelling, nutrient availability, and phytoplankton abundance all have potential influences on the population of planktic foraminifera. Planktic foraminifera occupy a range of trophic modes, operating as either functional mixotrophs, or exclusive heterotrophs, preying on phytoplankton, zooplankton, and detritus (Lipps and Valentine, 1970; Bé et al., 1977; Schiebel and Hemleben, 2017; Stoecker
et al., 2017; Fehrenbacher et al., 2018;). The seasonal and interannual variability of foraminiferal abundance is especially important to consider for applications in the paleontological record. For example, if a species is most abundant in the spring upwelling season, then geochemical records of that species will be skewed towards the spring rather than representing an annual average. To most effectively use foraminifera as proxies for past environmental conditions, it is necessary to consider and understand their modern, highly variable habitat at multiple sub-geologic timescales.




Ocean acidification resulting from anthropogenic $CO_2$ emissions is especially relevant in upwelling regions like the SBB (Gruber, 2011; Gruber et al., 2012; Hauri et al., 2009, 2013). Vertical water transport rates during the upwelling season have increased between the late 20th and early 21st centuries near SBB (García-Reyes and Largier, 2010; Jacox et al., 2018). Increased upwelling brings more nutrients to the surface, promoting primary production and decreasing the pH of surface water, potentially harming calcifying organisms that are sensitive to acidification, including foraminifera. Modern (2016) pteropods, for example, produce thinner aragonite shells in the more acidic, nearshore upwelling zones of the California coast compared to offshore, due to a decrease in calcification (Mekkes et al., 2021). Foraminifera also calcify thinner shells in response to ocean acidification (De Moel et al., 2009; Moy et al., 2009; Osborne et al., 2016; Pallacks et al., 2023).

The SBB sediment trap, located off the coast of California, USA, has provided a nearly continuous, high-resolution record of foraminifera shell and particle flux to the seafloor since 1993. Previous studies have analyzed the flux and species contributions of foraminifera to the sediment trap between 1993 and 1998 (Kincaid et al., 2000; Black et al., 2001), but this has not been revisited over the subsequent decades of rapid climate change. Paired with contemporaneous environmental data, we investigate the drivers of foraminiferal flux and species abundance in samples collected at biweekly intervals between 2014 and 2021 and compare our results to observations between 1993 and 1998 at the same site. Spanning nearly 30 years of rapid climate change, the SBB sediment trap series provides a unique opportunity to assess rates of change and add nuance to the underlying drivers of species composition and flux. In doing so we will be better able to place modern climate change in the context of past climate history.

## 1.1 Setting: Santa Barbara Basin, California

The SBB sediment trap is located approximately 32 km off the coast of Santa Barbara, California (Fig. 1). Due to the sedimentology of the basin, SBB is the location of several important climate archives (Kennett and Ingram, 1995; Behl and Kennett, 1996; Hendy and Kennett, 2000; Hendy et al., 2004; Field et al., 2006; White et al., 2013). On the floor of the basin, dark, lithogenic (mostly clay) layers are formed throughout the fall and winter. A light layer composed of biogenic silica (mostly diatoms) is deposited during the productive spring season (Thunell et al., 1995). The finely laminated sediment preserved in low-oxygen conditions provides an optimal location for palaeoceanographic studies.

Changing winds and ocean currents create strong seasonal cycles in SBB (Thunell, 1998). The California Current is the dominant source of water, transporting cold, nutrient and oxygen-rich water equatorward from the North Pacific (Bograd et al., 2001). The main current is located 500-800 km offshore (Auad et al., 2011). In the spring and summer, winds blowing southward along the coast promote upwelling off Point Conception, to the west of the basin (Thunell, 1998; Fiedler and Talley, 2006; Catlett et al., 2021). Near-surface oxygen, nitrate and chlorophyll concentrations are highest in the spring when upwelling occurs and stratification is low (Bograd et al., 2001; Black et al., 2011; Catlett et al., 2021; Simons and Catlett, 2023). Diatoms are the most abundant phytoplankton during the spring upwelling season, while picophytoplankton are more



abundant in the summer (Catlett et al., 2021). In the fall, winds relax and change direction, reducing upwelling. Stratification

increases and nutrient transport to the surface decreases (Bograd et al., 2001; Black et al., 2011; Catlett et al., 2021; Simons and Catlett, 2023). Eastern Tropical North Pacific water is transported to the basin via the inshore Counter Current and California Undercurrent (Fiedler and Talley, 2006; Auad et al., 2011; Davis et al., 2019; Alfken et al., 2021). The California Undercurrent brings warmer, saline, low-oxygen water to the region during the fall and winter (Bograd et al., 2001). Near-surface temperatures are coolest between April-May and warmest in September-October (Pak et al., 2004; SCB Marine

Biodiversity Observation Network, 2023).

In addition to the seasonal upwelling regime, conditions in SBB are impacted by Pacific multi-annual El Niño Southern Oscillation (ENSO) driven changes, the North Pacific Gyre Oscillation (NPGO), the Pacific Decadal Oscillation (PDO), and anthropogenically forced climate change (Black et al., 2001; Pak et al., 2004; Field et al., 2006). Peaks in dinoflagellate

abundance are associated with increased poleward flow, reduced upwelling, and the warm phase of the NPGO (Catlett et al., 2021). The warm phase of the PDO is also associated with reduced upwelling and increased poleward flow in SBB (Catlett et al., 2021). The northward advection of relatively warm water during the winter is amplified during some El Niño events (Lynn and Bograd, 2002; Auth et al., 2015). From 1993 to 2021, two strong El Niño events occurred in 1997/1998 and 2015/2016 with weak El Niño conditions occurring in 1993/1994 and 2019 (Trenberth et al., 2024). The 1997/1998 and 2015/2016 El

Niño events both exhibited low phytoplankton biomass compared to other years (Catlett et al., 2021). Kincaid et al. (2000) found lower foraminiferal flux during the weak El Niño conditions of 1993/1994. Conversely, Black et al. (2001) reported foraminiferal flux values that were four times higher during the upwelling season in 1997 compared to 1996, with an increase in the relative abundance and flux of *G. bulloides* during the El Niño year 1997.





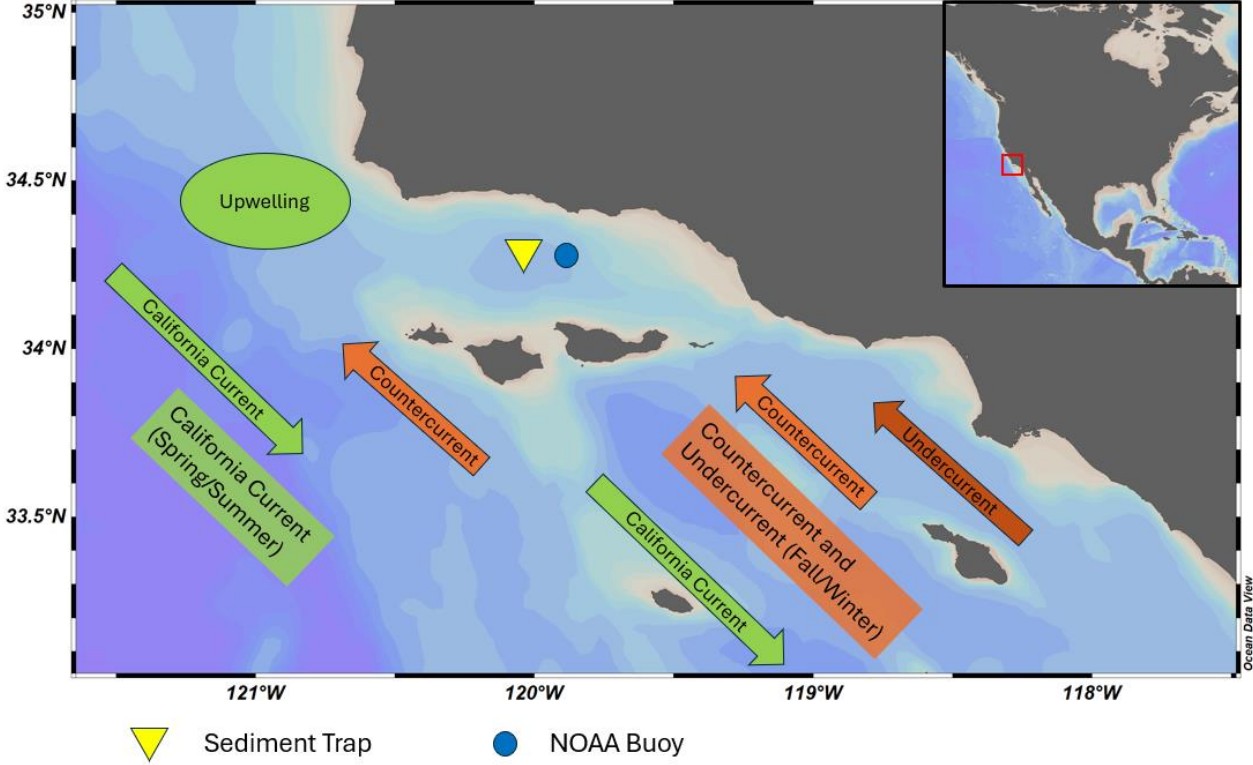


**Figure 1: Map of Santa Barbara Basin. Major seasonal current directions are shown for spring/summer (green) and fall/winter (orange). The sediment trap and NOAA buoy 46053 are represented by the yellow triangle and blue circle, respectively. Schlitzer, R., Ocean Data View, https://odv.awi.de, 2018.**

## 2   Methods

**2.1 Sample collection, wet picking and assemblages**

Samples were collected at 10-to-14-day intervals over ~ 6-month intervals using a moored McLane Parflux 78H sediment trap deployed at depths > 400 m in the central SBB, CA since 1993 (34°14'42.14'' N, 120°03'33.45'' W, water column depth of ~589 m; Eichhubl et al., 2001). Samples were preserved in a borate-buffered formalin solution and split, with a 1/16th split used for foraminiferal flux and species counts, excluding July-October 2015 and May-November 2020.


Each sample was wet picked for foraminifera following the procedure outlined in Kincaid et al. (2000). The formalin solution was removed using a tap water rinse and samples sieved using a 125µm sieve, consistent with previous SBB assemblage studies (Kincaid et al., 2000; Black et al., 2001). All foraminifera present in the sample were separated from other sediment





trap material using a fine paintbrush and allowed to air dry. Following wet picking, foraminifera were sorted by species and
counted to determine foraminiferal flux and relative abundance.

Foraminiferal flux was calculated from the species counts (1/16th splits) to determine the #foraminifera $m^{-2}$ $day^{-1}$ for each full
sample. This method is consistent with the flux calculations used in Kincaid et al. (2000) and Black et al. (2001) for the same
sediment trap with an area of 0.5 $m^2$. Possible sources of error include variability associated with creating a $1/16^{th}$ split with a
wheel splitting device, the potential for missing foraminifera during wet picking, and infrequent breakage of foraminiferal
tests, the latter two sources resulting in possible under-counting of foraminifera. The error in foraminiferal flux data from this
study is likely similar to that of Kincaid et al. (2000) and Black et al. (2001), as both the methods and sediment trap are
consistent.

## 2.2 Additional Datasets

Environmental data was collected from a variety of additional datasets. A description of each dataset and the data obtained
from each source is provided in Table A1. All environmental data apart from CTD casts are surface measurements (< 5m),
and data was filtered to include only measurements from SBB between 2014 and 2021.

## 2.3 Statistical Methods

We used a correlation matrix to initially investigate relationships between every species and environmental variable. The
matrix uses Pearson correlation and the Benjami and Hochberg (1995) False Discovery Rate (FDR) p-value correction for
multiple-hypothesis significance testing ($p < 0.05$) and was carried out in MATLAB using the 'fdr_bh' function (Groppe,
2024). Thermocline characteristics were calculated from Plumes and Blooms and CalCOFI CTD data using MATLAB (See
Table A1). The downcast portions of the CTD files were binned in 1m intervals, and the depth of the mixed layer, thermocline
depth, bottom of the thermocline, and thermocline slope were calculated. The thermocline depth was calculated as the depth
over which the temperature change was the greatest. Thermocline slope was calculated as the average slope of the thermocline
between the bottom of the mixed layer and bottom of the thermocline. Canonical Correlation Analysis (CCA) provided further
analysis of the relationships between species and environmental data and was conducted using the 'vegan' package in RStudio
(Oksanen et al., 2024; Posit Team, 2024). We used a periodic regression model to determine the multi-year seasonal
distribution for each species in our dataset and performed the same periodic regression analysis on species data from Kincaid
et al. (2000) and Black et al. (2001) to provide a comparison between 1993-1998 and 2014-2021. The model uses the $log_{10}$ of
foraminiferal flux and the periodic date transformation outlined in Jonkers and Kucera (2015), where the transformed date for
each sample is equal to day of the year divided by 365.25 and multiplied by $2\pi$. Data visualization for the periodic regression
was carried out with the 'ggplot2' package in RStudio (Wickham, 2009; Posit Team, 2024). Periodic regressions for each
timeseries and species are shown, e.g., the log of species flux is plotted against day of the year. To compare the seasonal
distribution of each species between 1993-1998 and 2014-2021, species fluxes were binned by month for the duration of each





timeseries, and the average flux values were calculated for each month. A first order polynomial linear regression (in MATLAB) was used to assess rates of change in foraminiferal flux between the two sample sets. A date transformation was applied to each sample such that the new date is equal to years elapsed since the collection of the first sediment trap sample in 1993. This date transformation results in a best fit line with a slope in units of foraminiferal flux per year. Two-sided Wilcoxon

rank sum tests were used to test the significance of changes in median foraminiferal flux. An unequal variance t-test was used to assess the significance of changes in mean PIC/POC.

## 3    Results

### 3.1 The Santa Barbara Basin sediment trap: 2014-2021

This study presents results from 164 sediment trap samples, collected between May 24, 2014, and November 11, 2021. The

total number of 60,006 individual foraminifera includes 25 morphospecies (Tables 1 and A2). Images of the 10 most abundant morphospecies are presented in Fig. A1. The most abundant species are *G. bulloides*, *T. quinqueloba*, and *N. incompta* (Table 1). *Globigerina bulloides* is consistently abundant throughout the year, with some slight seasonal peaks in the spring and summer (Figs. 2, 3, and 4). *Turborotalita quinqueloba* is present year-round, but most abundant during the spring and summer when upwelling is active (Figs. 2, 3, and 4). *Neogloboquadrina incompta* increases in abundance during the fall and winter

months (Figs. 2, 3, and 4). *Globoturborotalita rubescens* and *Globigerinoides ruber* are the most abundant subtropical species, followed by *Orbulina universa* and *Neogloboquadrina dutertrei* (Table 1). Some rare species (defined here as species contributing less than 0.7% of the total assemblage) were also found, including *Globorotalia truncatulinoides* and the tropical *Trilobatus sacculifer* (Table 1).

Foraminiferal flux throughout the study period averages 850 foraminifera m$^{-2}$ day$^{-1}$ throughout the study period. The maximum total flux of 3,895 foraminifera m$^{-2}$ day$^{-1}$ occurs in March 2019 (Fig. 2). Foraminiferal species *G. bulloides* and *T. quinqueloba* each dominated total fluxes depending on the sample and season (Fig. 2). *Neogloboquadrina incompta* comprises a greater proportion of total flux when the abundances of *G. bulloides* and *T. quinqueloba* are low, most commonly in the fall months (Fig. 2). *Globigerinita glutinata* and *G. scitula* are present throughout the year with lower fluxes than the three major species

(Table 1 and Fig. 2). Subtropical species (*G. ruber*, *G. rubescens*, *O. universa*, and *N. dutertrei*) have relatively low fluxes that peak in the fall and winter when upwelling is reduced (Figs. 2, 3, and 4). During the upwelling season, the flux of subtropical species is near zero (Figs. 2, 3, and 4). Exceptions to this general pattern occur in late summer/early fall of 2014, when fluxes are especially high (Fig. 2). The flux of *G. rubescens* is also high in the late fall of 2015 (Fig. 2).




| Species | Count | Species | Count |
|---|---|---|---|
| G. bulloides | 20,957 | B. digitata | 77 |
| T. quinqueloba | 16,545 | G. calida | 71 |
| N. incompta | 13,486 | G. siphonifera | 34 |
| G. glutinata | 2.489 | B. variabilis | 34 |
| G. scitula | 961 | T. parkerae | 23 |
| G. rubescens | 828 | G. hexagonus | 20 |
| G. ruber | 596 | T. iota | 18 |
| other | 572 | G. inflata | 12 |
| O. universa | 539 | G. truncatulinoides | 12 |
| N. dutertrei | 493 | T. sacculifer | 7 |
| G. falconensis | 461 | H. pelagica | 3 |
| G. uvula | 365 | G. hirsuta | 1 |
| N. pachyderma | 164 | | |
| G. tenellus | 98 | Total | 60,006 |


**Table 1**: All species counts (from 1/16$^{th}$ splits) in order of abundance.






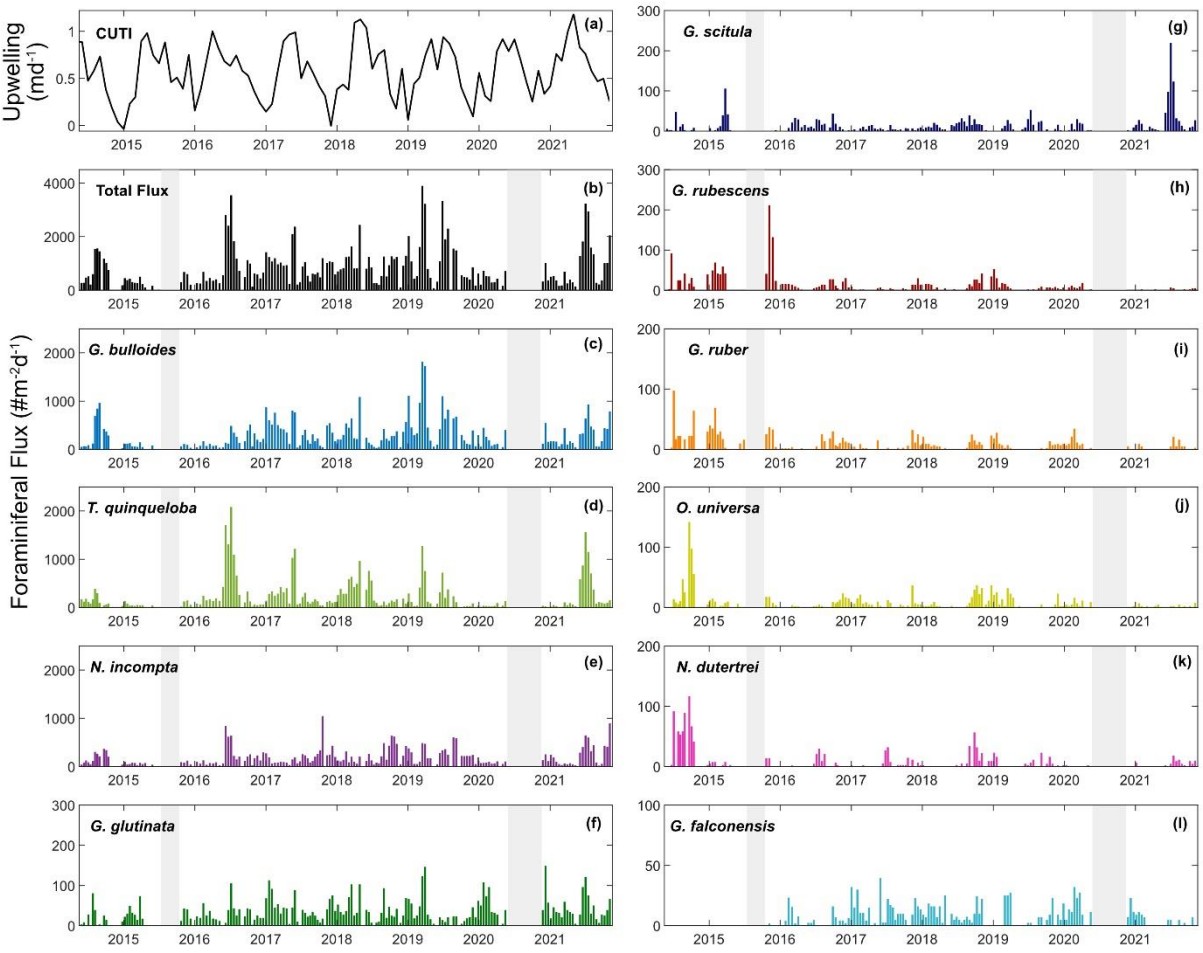

**Figure 2: Timeseries of upwelling (CUTI), total foraminiferal flux, and the 10 most abundant species presented in order of abundance. Transitional to subpolar species (c-g and l) are represented in cool colors, and subtropical species (h-k) are represented in warm colors. Gray bars indicate periods without sample collection.**





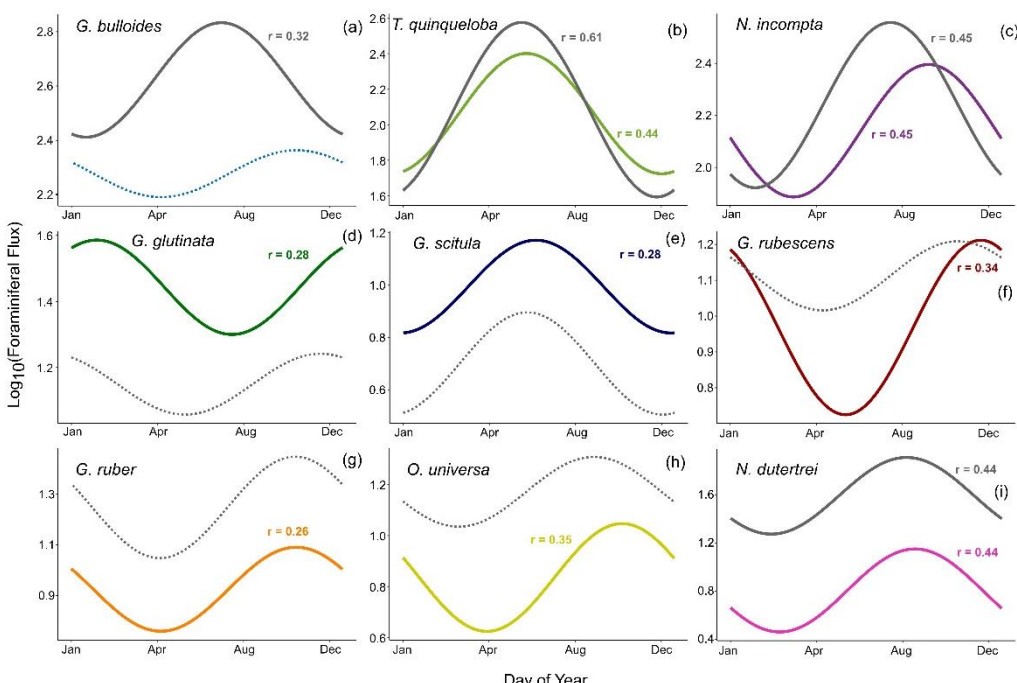

**Figure 3: Periodic regression of the nine most abundant species presented in order of abundance. Model fits from 1993-1998 are shown in gray (data from Kincaid et al. 2000 and Black et al. 2001) and fits from 2014-2021 are shown in color. Solid lines show significant models where p < 0.05. Non-significant models (p > 0.05) are dotted.**





**Figure 4: Seasonal species plots from 1993-1998 (gray) (data from Kincaid et al. 2000 and Black et al. 2001) and 2014-2021 (this paper) (colors are overlain). CUTI (a) is shown following the same binning and averaging method as species data, along with total foraminiferal flux (b). Species chosen include the most abundant species (c-l) and rare species**
**(m-p) that were consistently speciated throughout both time periods (*B. digitata*, *G. calida*, *G. siphonifera*, and *T. sacculifer*), presented in order of abundance.**

Correlations (r) between each species and environmental variable are shown in Fig. 5. Surface measurements are presented
with a 14-day lag to account for the growth and life cycle of foraminifera in response to water-column environmental conditions. Once foraminifera die, they sink to the depth of the sediment trap in 1-4 days (Takahashi and Be, 1984). Variables from the SBB sediment trap are not lagged.

Total foraminiferal flux is positively correlated with sample mass and other parameters from the sediment trap, with larger
samples containing both more foraminifera and more particle flux overall (Fig. 5). *Neogloboquadrina incompta* and *T. quinqueloba* are generally positively correlated with sediment trap parameters as well (Fig. 5). The most abundant species, *G. bulloides*, present in high abundance in SBB throughout every season and every year, has no significant correlations with environmental parameters (Figs. 2, 4, and 5). *Turborotalita quinqueloba* has the strongest correlation with upwelling (r = 0.251, p < 0.05) and sediment trap opal (r = 0.445, p <0.05) out of every species (Fig. 5). *Neogloboquadrina incompta* correlates
positively with SST (r = 0.336, p < 0.05), picophytoplankton (r = 0.323, p < 0.05), and prymnesiophytes (r = 0.281, p < 0.05) (Fig. 5). This is consistent with *N. incompta's* seasonal pattern, peaking in late September with generally higher abundance in the latter half of the year (Figs. 3, 4, and 5). *Globorotalia scitula* has similarities to *T. quinqueloba*, with an especially strong negative correlation (r = -0.846, p < 0.05) to the thermocline slope (Fig. 5). *Turborotalita quinqueloba* has a negative correlation to the depth of the bottom of the thermocline as well (r = -0.431, p < 0.05) with abundances higher when the
thermocline is shallower (Fig. 5). These species seem to prefer the steep, shallow thermocline characteristic of the upwelling season. The subtropical species show an opposite trend, where *G. ruber* is positively correlated with a deeper mixed layer (r = 0.356, p < 0.05) (Fig. 5). When the mixed layer is deeper, the thermocline is more gently sloping and extends deeper. This condition correlates significantly with *G. falconensis* (r = 0.328, p < 0.05) (Fig. 5). Other subtropical species, *O. universa* and *N. dutertrei*, are significantly correlated to SST (r = 0.350, p < 0.05 and r = 0.449, p < 0.05, respectively) rather than thermocline
characteristics, preferring warmer water (Fig. 5).

Correlations for rare species contributing less than 0.7% of the total assemblage were run separately from the more abundant species due to less overlap between species occurrence and environmental data. Species that contribute less than 0.002% of the total assemblage were excluded. Rare species have fewer significant correlations with environmental parameters compared
to the abundant species (Fig. A2). There are no significant correlations (p < 0.05) with sample mass and rare species (Fig. A2).





Much like *N. incompta*, *N. pachyderma* has a positive correlation with picophytoplankton (r = 0.472, p < 0.05) (Fig. A2). *Globigerinella siphonifera*, a subtropical species, has a positive correlation with SST (r = 0.346, p < 0.05) (Fig. A2). *Globorotaloides hexagonus* has a negative correlation with thermocline slope (r = -0.680, p < 0.05), consistent with *T. quinqueloba* and *G. scitula* (Fig. A2).


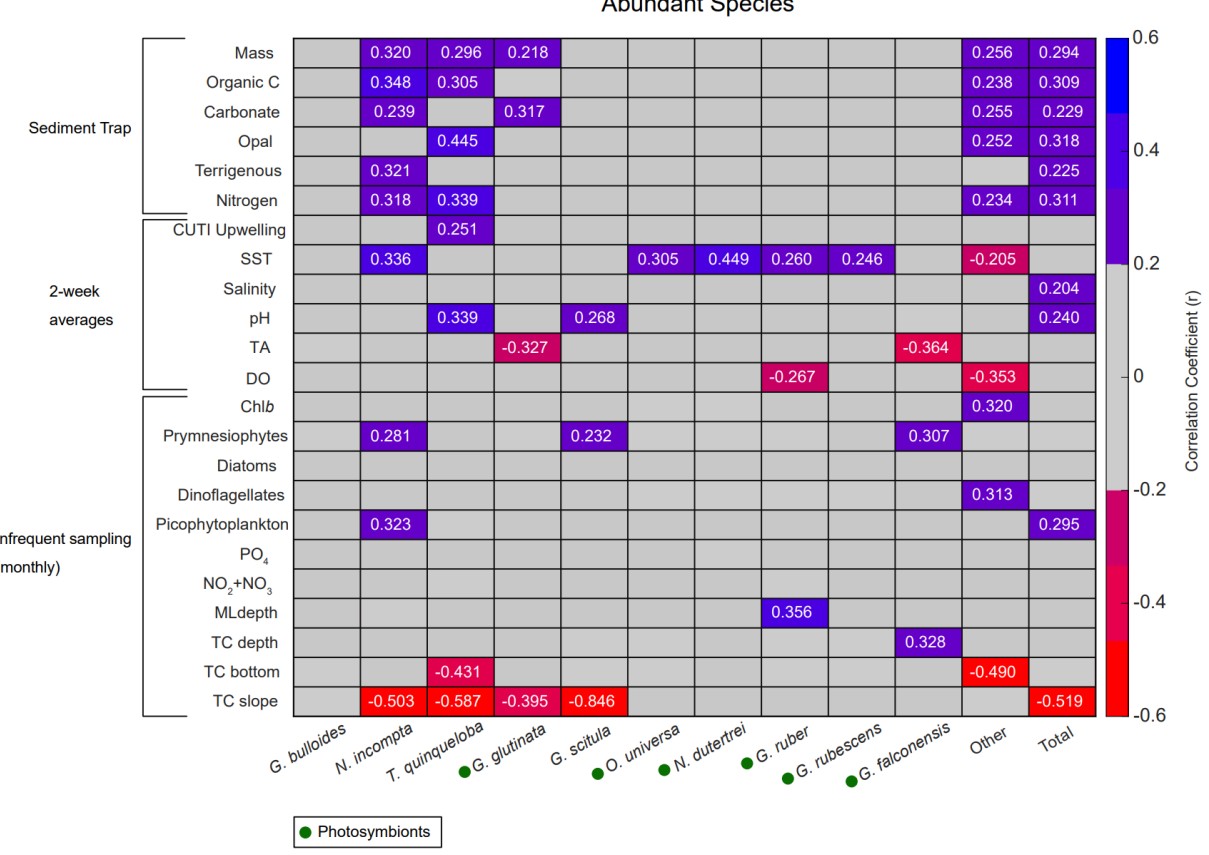

**Figure 5: Correlation matrix of abundant species flux (x axis) with environmental parameters (y axis). Pearson correlation coefficients (r) for statistically significant correlations (p < 0.05) are listed as numbers. Blue shades represent positive correlations, and red shades represent negative correlations. Gray represents correlation coefficients < 0.2. Presence of photosymbionts based on Takagi et al. (2019), Schiebel and Hemleben (2017), and Meilland et al. (2022) is represented by green circles.**





### 3.1.1 Canonical Correlation Analysis

CCA1 constrains 21.7% of variability and is primarily driven by SST and surface dissolved oxygen in the positive direction. Salinity, pH, and CUTI drive CCA1 in the negative direction, along with fluxes of nitrogen, opal, and organic carbon from the

sediment trap (Fig. 6 and Table 2). *Turborotalita quinqueloba* is most closely associated with variables negatively loaded on CCA1 compared to other species (Fig. 6 and Table 2). *Globorotalia scitula* is also negatively loaded to CCA1. *Globigerina bulloides* has a positive loading on CCA1 and trends in the opposite direction from upwelling related variables and *T. quinqueloba* (Fig. 6 and Table 2). CCA2 constrains 15.0% of variability. SST and mixed layer depth are positively loaded on CCA2, along with the variables in the northwest quadrant (CUTI, pH, opal, nitrogen, salinity, organic carbon) (Fig. 6). The

species most associated with SST are *O. universa*, *G. ruber*, *N. dutertrei*, and *G. rubescens* (Fig. 6). *Globoturborotalita rubescens* has the strongest positive loading on CCA2, positively associated with SST and mixed layer depth and negatively with surface dissolved oxygen (Fig. 6 and Table 2). Loadings for each species and environmental variable are listed in Table 2.


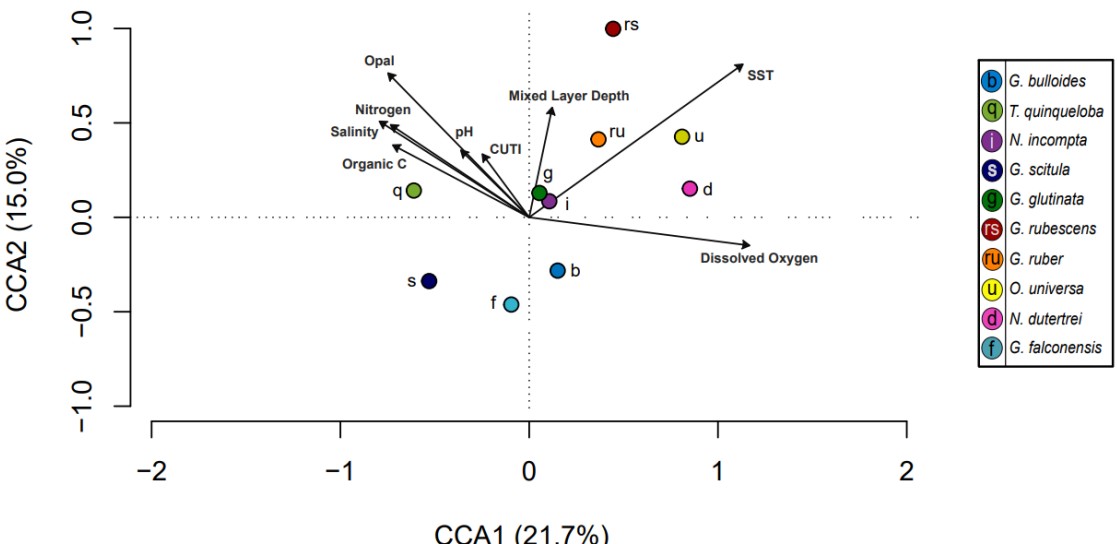

**Figure 6: Canonical Correlation Analysis of 10 most abundant species (points) and environmental variables (arrows). Environmental variables include sediment trap parameters (organic carbon, nitrogen, opal) and water column parameters (salinity, surface pH, Coastal Upwelling Transport Index (CUTI), mixed layer depth, sea surface**

**temperature, surface dissolved oxygen). The full model constrains 54.7% of variability, with CCA1 and CCA2 accounting for a combined 36.7%. Adjusted $R^2$ = 0.22 and p = 0.016.**






| Species | Loading | | Environmental | Loading | |
|---|---|---|---|---|---|
| | CCA1 | CCA2 | | CCA1 | CCA2 |
| *G. bulloides* | 0.151 | -0.282 | Organic carbon (OC) | -0.405 | 0.214 |
| *N. incompta* | 0.106 | 0.085 | Opal | -0.419 | 0.428 |
| *T. quinqueloba* | -0.611 | 0.142 | Nitrogen (N) | -0.412 | 0.275 |
| *G. glutinata* | 0.055 | 0.129 | CUTI upwelling | -0.139 | 0.187 |
| *G. scitula* | -0.530 | -0.338 | Sea surface temperature (SST) | 0.635 | 0.455 |
| *O. universa* | 0.810 | 0.427 | Salinity (Sal) | -0.445 | 0.285 |
| *N. dutertrei* | 0.851 | 0.152 | pH | -0.203 | 0.199 |
| *G. ruber* | 0.367 | 0.413 | Dissolved oxygen (DO) | 0.655 | -0.084 |
| *G. rubescens* | 0.445 | 0.998 | Mixed layer depth (ML) | 0.068 | 0.326 |
| *G. falconensis* | -0.095 | -0.462 | | | |


**Table 2: Loadings for species and environmental variables on CCA1 and CCA2.**

### 3.2 Seasonal patterns in foraminiferal flux and comparison to 1993-1998

Total foraminiferal flux between 2014-2021 has decreased relative to 1993-1998. The median of the total flux, a metric chosen

to avoid biasing by the single-sample 1997 peak in flux, decreased by 372.5 foraminifera $m^{-2}d^{-1}$, or 37.9 % ($p = 3.8 \times 10^{-5}$). Median *G. bulloides* flux decreased by 267.7 foraminifera $m^{-2}d^{-1}$ (-58.8%, $p = 1.4 \times 10^{-8}$), and *N. incompta* flux decreased by 84.3 foraminifera $m^{-2}d^{-1}$ (-40.5%, $p = 0.003$). The largest contributor to the decrease in total foraminiferal flux is *G. bulloides*, followed by *N. incompta*. *Turborotalita quinqueloba* flux remained unchanged relative to 1993-1998 ($p = 0.74$) (Fig. 8). Note that all changes in median foraminiferal flux remain significant at the $p < 0.05$ level even after the removal of the peak flux

that occurred in 1997.





The rates of change of foraminiferal flux for all comparable and abundant species are based on bi-weekly fluxes from 1993-2021 and are presented in Table 3, where negative values indicate a decrease in flux between 1993 and 2021 (Tables 3 and A3, Fig, A3). An additional table of rates of change with the peak flux point in 1997 removed is presented in Table A3.

Between 1993 and 2021, the change in total foraminiferal flux per year was -22.26 (Table 3). *Globigerina bulloides* had the highest rate of change, with foraminiferal flux decreasing by 17.07 per year (Table 3 and Fig. A3). Subtropical species flux decreased over time, with changes between -2.45 (*N. dutertrei*) and -0.32 (*G. rubescens*) foraminiferal flux per year. However, the median flux of G. rubescens did not change significantly from 1993-2021 (p = 0.74). Only two species, *G. glutinata* and *G. scitula*, exhibited positive rates of change (Table 3).


This timeseries contains two strong El Niño events. There was a large total flux peak in the summer of 1997, during the 1997/1998 El Niño (Fig. 7; Black et al., 2001), where the three most common species reached their maximum flux values. *Globigerina bulloides* contributed most to total foraminiferal flux, followed by *T. quinqueloba* (Fig. 7; Black et al., 2001). The 2015/2016 El Niño does not exhibit the same pattern. *Globigerina bulloides* abundance is extremely low throughout the event,

while *T. quinqueloba* and *N. incompta* abundance increase during the summer of 2016, with fluxes that surpass the summer peaks of every other year from 2014-2021 (Fig. 7). Subtropical species flux increased during both strong El Niño events (Fig. 7; Black et al., 2001). Fluxes of *G. ruber* and *N. dutertrei* are consistently lower from 2014-2021 compared to 1993-1998 (Fig. 7; Kincaid et al., 2000; Black et al., 2001). *Orbulina universa* flux is high in 2014, but it is much lower than it was in the 1990s in every subsequent year (Fig. 7). From 1993-1997, *G. rubescens* flux was very low, but increased to reach its peak value

during the 1997/1998 El Niño (Fig. 7; Black et al., 2001). From 2014-2019, *G. rubescens* flux is higher than it was from 1993-1997 and decreases from 2020-2021 (Fig. 7).






Figure 7: Timeseries of CUTI (a) and SBB sediment trap foraminiferal flux from Kincaid et al. (2000) and Black et al. (2001) (left) and this study (right). Total flux (b) and the three most abundant species, *G. bulloides* (c), *T. quinqueloba* (d), and *N. incompta* (e). Major subtropical species are shown in warm colors: *G. rubescens* (f), *G. ruber* (g), *O. universa* (h), and *N. dutertrei* (i). Gray shading indicates periods without sample collection.





| Species | Rate of change (foraminiferal flux per year) | r |
|---|---|---|
| Total | -22.26 | 0.20 |
| *G. bulloides* | -17.07 | 0.27 |
| *T. quinqueloba* | -0.04 | 0.001 |
| *N. incompta* | -3.38 | 0.17 |
| *G. glutinata* | 0.82 | 0.32 |
| *G. scitula* | 0.55 | 0.25 |
| *G. rubescens* | -0.32 | 0.12 |
| *G. ruber* | -0.93 | 0.38 |
| *O. universa* | -0.58 | 0.33 |
| *N. dutertrei* | -2.45 | 0.51 |

**Table 3: Rates of change from 1993-2021 presented as foraminiferal flux per year (2nd column) and correlation coefficients (r) of the linear regression for each species (3rd column).**

In a single year, *G. bulloides* flux fluctuates noticeably and exhibits high interannual variability (Fig. 2). However, the 2014-2021 monthly averages show no clear seasonal peak, and the periodic regression of *G. bulloides* is not significant ($p > 0.05$) (Figs. 3 and 4). There is a significant periodic regression for *G. bulloides* from 1993-1998, that shows higher flux in May and a peak in late July, likely influenced by a single high-flux sample during the 1997 El Niño in May and an inshore meander of the California Current during November of 1996 (Black et al., 2001; Fig. 3).

The median flux of *T. quinqueloba* has remained consistent (+1%) since the 1990s. From 2014-2021, there is a seasonal peak in June and July, consistent with the upwelling season (Fig. 4). The highest flux occurred in May from 1993-1998, about one month earlier (Fig. 4). This shift in timing is less evident in the periodic regression of *T. quinqueloba*, where the timing of peak flux has remained similar (Fig. 3).

The flux of *N. incompta* has decreased since the 1990s (Figs. 2, 3, and 4). The species is most abundant from June to December, with peak flux in the fall (Figs. 3, 4, and 9). The largest decrease in flux occurred between May and September (Fig. 4). The seasonal distribution pattern looks quite similar, but the seasonal pattern has shifted by 52 days later compared to the 1990s, with peak flux occurring in late September instead of early August (Fig. 3).



Three species, *G. glutinata*, *G. scitula*, and *B. digitata* have consistently increased in flux since the 1990s (Fig. 4). Much like *T. quinqueloba*, *G. scitula* is most abundant in July, but it is less abundant throughout the rest of the year (Fig. 4). *Globorotalia scitula* has a more distinct seasonal pattern than *G. glutinata*, as it is present in highest abundance during the upwelling season (Figs. 3 and 4). The seasonal patterns of *G. glutinata* and *G. falconensis* are similar, with highest fluxes from January to March and lower abundances during the upwelling season (Fig. 4). The species counts from 1993-1998 do not include *G. falconensis*,

so there is no comparison to the 1990s.

The flux of subtropical species, *G. ruber*, *O. universa*, and *N. dutertrei* have greatly decreased over time, with changes in median flux by 12.1, 11.5, and 45.7 foraminifera m$^{-2}$ d$^{-1}$, respectively (all $p < 0.05$) (Figs. 4 and 7). Their seasonal distributions have remained similar (Fig. 4). They are most abundant during non-upwelling in the winter and fall (Figs. 4 and 9). Unlike the

other subtropical species, the median flux of *G. rubescens* has not decreased significantly ($p > 0.05$). *Neogloboquadrina dutertrei* shows a seasonal pattern similar to *N. incompta*, where flux is low from January to April and increases in May (Figs. 3 and 4). This seasonal pattern is distinct between 1993-1998, but less so from 2014-2021 due to a significant decrease in flux (Figs. 3 and 4). The median fluxes of subtropical and tropical species, *G. siphonifera*, *G. calida,* and *T. sacculifer* have decreased since the 1990s (all $p < 0.05$), such that only 34, 71, and 12 individuals, respectively, counted from 2014-2021

(Table 1 and Fig. 2).

### 3.3 Relative abundance

Moving from the 1990s to 2010s, the relative abundance of *G. bulloides* has decreased by 16% and *T. quinqueloba* has increased by 11% (Fig. 8). The relative abundance of *N. incompta* has increased slightly (3%) (Fig. 8). Subtropical species have decreased by 6%, contributing only 3% of the total assemblage from 2014-2021 (Fig. 8). This decrease has been countered

by an increase in other species categories (*G. glutinata*, *G. scitula*, *G. falconensis*, 'other') (Fig. 8). The relative abundance of each species varies seasonally (Fig. 9). *Globigerina bulloides* is most abundant in the winter (43%) and spring (41%), with moderate relative abundance in the summer (28%) and fall (35%) (Fig. 9). *Turborotalita quinqueloba* is most abundant in the spring (35%) and summer (39%), with especially low abundance in the fall (11%) and winter (16%) (Fig. 9). *Neogloboquadrina incompta* is most abundant in the fall (38%), with moderate abundance in the winter (23%) and summer (23%), and low abundance in the spring (12%) (Fig. 9).

(23%), and low abundance in the spring (12%) (Fig. 9).



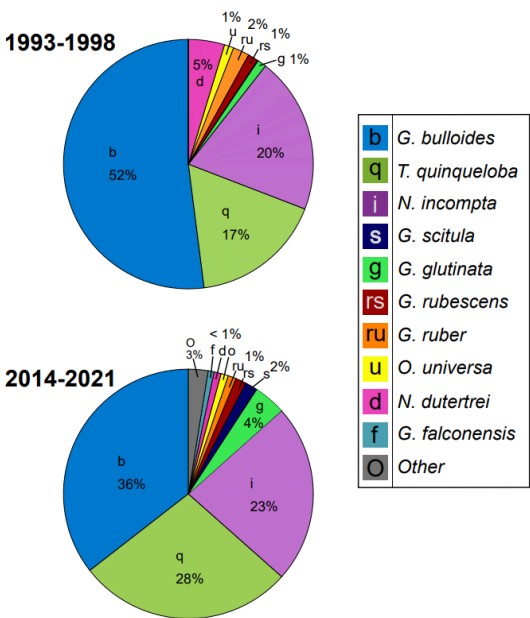

**Figure 8: Relative abundance of foraminiferal species from 1993-1998 (data from Kincaid et al., 2000 and Black et al., 2001) and 2014-2021.**

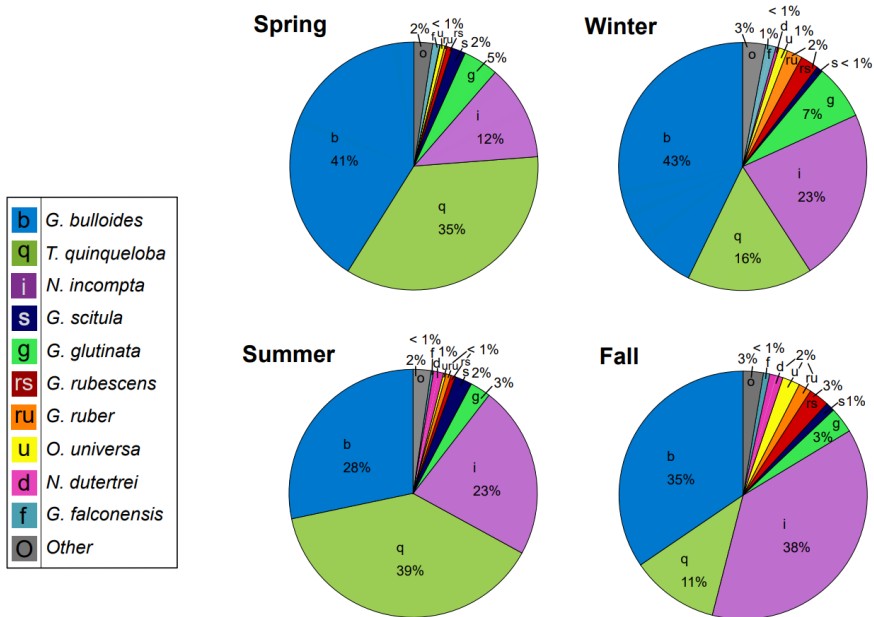

**Figure 9: Seasonal relative abundance of planktic foraminifera from 2014-2021. Winter (December, January, February), spring (March, April, May), summer (June, July, August), and fall (September, October, November).**



## 4    Discussion

### 4.1    Ocean acidification as a driver of *G. bulloides* flux decrease

Ocean acidification, due to oceanic uptake of anthropogenic $CO_2$, is a global issue, and the impacts are exacerbated within eastern boundary upwelling systems such as the California Current system (Feely et al., 2008; Gruber, 2011; Gruber et al., 2012; Hauri et al., 2013; Bograd et al., 2023). Calcite and aragonite saturation states and pH within the California eastern boundary upwelling system are decreasing, and corrosive water (aragonite saturation < 1 and pH < 7.75) has been recorded as shallow as 40-120m depth along the California coast (Feely et al., 2008; Gruber et al., 2012; Bograd et al., 2023). The saturation horizon is projected to continue to shoal (Gruber et al., 2012). Based on the Coastal Upwelling Transport Index, upwelling near SBB has increased in both duration and magnitude relative to 1993-1998 (Jacox et al., 2018; Figs. 4 and 7). This is consistent with the overall trend of increased upwelling along the California coast from 1982-2008 (García-Reyes and Largier, 2010). Average monthly vertical water transport rates only reached 0.6 m day$^{-1}$ or above in April and June from 1993-1998 (Jacox et al., 2018; Fig. 4). In contrast, between 2014 and 2021, average monthly vertical water transport rates consistently reached 0.6 m day$^{-1}$ or above from April through September (Jacox et al., 2018; Fig. 4). This extends the exposure time of foraminifera to low pH upwelled waters by two to three months.

The decrease in median *G. bulloides* flux (-58.8%) between 1993-1998 and 2014-2021 is the biggest driver in the decline in total foraminiferal flux (-37.9%) (Fig. 7). *Globigerina bulloides* continues to inhabit SBB throughout the year with a high degree of interannual variability, showing a lack of seasonal peaks when averaged over multiple years (Kincaid et al., 2000; Black et al., 2001; Figs. 2, 3, and 4). This species is further positively associated with surface dissolved oxygen, which is largely an indicator of surface primary productivity (Fig. 6). As *G. bulloides* is a heterotroph and opportunistic feeder, this relationship to primary productivity is expected (Schiebel and Hemleben, 2017). At the same time, *G. bulloides* is negatively associated with environmental upwelling variables, including CUTI, pH, organic carbon, nitrogen, and opal (Fig. 6).

Although *G. bulloides* inhabits upwelling regions and is the most abundant species of planktic foraminifera in SBB, it is also sensitive to ocean acidification in ways that affect shell thickness, survival, and reproduction (Davis et al., 2017; Osborne et al., 2020). Osborne et al. (2020) showed that area normalized shell weights of *G. bulloides* from the SBB sediment record have decreased throughout the 20th century in response to global ocean acidification and are modulated by changes in upwelling strength associated with the PDO (Osborne et al., 2020). In addition to calcification, the oxygen utilization and spine repair of *G. bulloides* decrease with decreasing pH in culture (Davis et al., 2017), which may impact the ability of *G. bulloides* to survive and reproduce. In the context of rapid acidification in the California Current system and an increase in local upwelling intensity, the decrease in *G. bulloides* flux is most likely driven by ocean acidification such that conditions within SBB are moving beyond the range of tolerable conditions for this species. Under a business-as-usual emissions scenario, ocean acidification





will intensify, possibly further reducing foraminifera presence and subsequent inorganic carbon flux to the seafloor (Orr et al., 2005; Hofmann and Schellnhuber, 2009; Beaufort et al., 2011; Meyer and Riebesell, 2015; Kiss et al., 2021).

**4. 2 Decrease in total foraminiferal flux**

Our record shows a 37.9% decrease in median total foraminiferal flux over a period of 28 years. Marine calcifiers such as planktic foraminifera and coccolithophorids comprise a significant proportion of calcium carbonate flux to the seafloor (Schiebel, 2002; Baumann et al., 2004; Langer, 2008; Tanhua et al., 2013). Their calcification and subsequent inorganic carbon burial are important components of the marine carbon cycle (Zondervan et al., 2001; Schiebel, 2002; Langer, 2008; Tanhua et al., 2013). Calcification removes carbonate ion, thereby reducing alkalinity and leading to a decrease in the pH of seawater

and the ability of the surface ocean to uptake atmospheric $CO_2$ (Frankignoulle et al., 1994). Thus, a decrease in calcification from planktic foraminifera has the potential to actually increase the ability of the ocean to sequester $CO_2$ and act as a negative feedback to anthropogenic $CO_2$ inputs (Zondervan et al., 2001; Hofmann and Schellnhuber, 2009; Tanhua et al., 2013; Meyer and Riebesell, 2015).

Between 1993-1998 and 2014-2021, the decrease in total foraminiferal flux contributed to a 21.9% decrease in total carbonate flux to the sediment trap, while organic carbon flux remained relatively stable (-0.6%) (Figs. 2 and 10). In essence, the removal of alkalinity from the surface ocean through calcification has been reduced while drawdown of $CO_2$ by phytoplankton through photosynthesis continues (Frankignoulle et al., 1994; Tanhua et al., 2013; Zondervan et al., 2001). This results in a reduction of the ratio of the flux of particulate inorganic carbon (PIC) to particulate organic carbon (POC) in the SBB sediment trap of

16.3% (p = 1.7 x $10^{-4}$), indicating a strengthening of the biological carbon pump and increasing potential for $CO_2$ sequestration in the deep ocean (Zondervan et al., 2001; Tanhua et al., 2013; Fig. 10). Although a decrease in calcification has negative consequences for calcifying organisms, it may already provide stabilizing feedback to atmospheric $CO_2$ inputs.



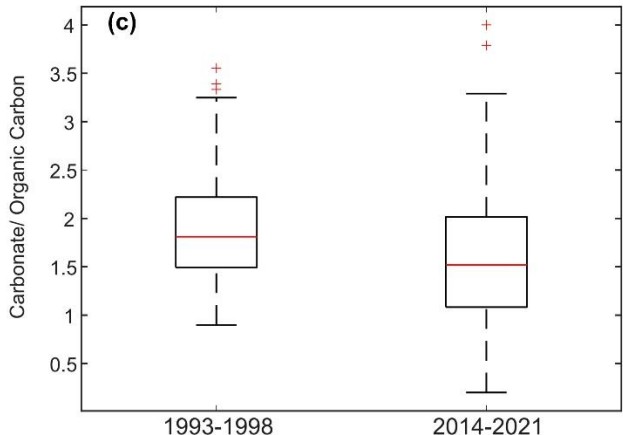

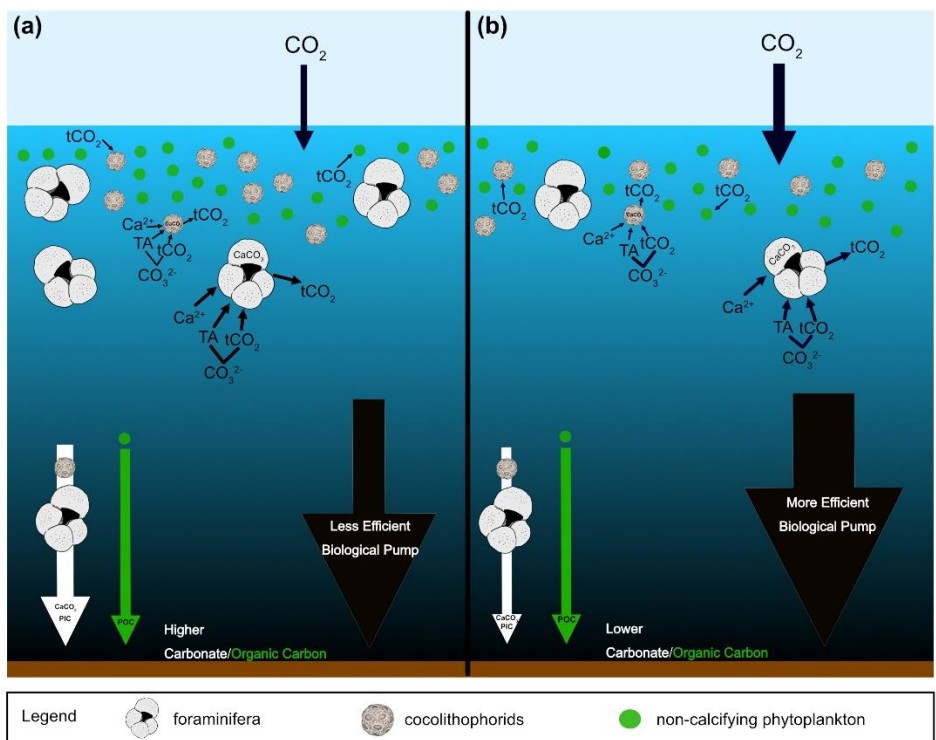

**Figure 10: Boxplot of PIC/ POC as carbonate/ organic carbon for two time periods, 1993-1998 and 2014-2021, (c). Schematic of a high PIC/ POC scenario (a), similar to the 1993-1998 median, with high calcification and a less efficient biological pump, resulting in less oceanic uptake of atmospheric $CO_2$. Schematic of a low PIC/POC scenario (b), similar to the 2014-2021 median, with less calcification and a more efficient biological pump, resulting in more oceanic uptake of atmospheric $CO_2$.**



**4.3 Decrease in subtropical species flux and relative abundance**

Subtropical species have decreased in both flux and relative abundance since the 1990s (Figs. 4, 7, and 8). In the context of a 0.38° C global SST increase between 1993 and 2021 due to anthropogenically-forced warming (NOAA, 2024), this is an
unexpected finding. Foraminiferal assemblages are shifting poleward globally in response to warming ocean temperatures (Jonkers et al., 2019), and the flux of subtropical species has increased throughout the warming 20th century in the SBB sediment record (Field et al., 2006). We argue that although the California upwelling system is warming over time, it is doing so more slowly than non-upwelling coastal zones, with the slowest warming rate closest to shore (Seabra et al., 2019; Bograd et al., 2023). While the rest of the coastal ocean warms and stratifies, the California system remains relatively cool with active
and increasing upwelling (García-Reyes and Largier, 2010; Seabra et al., 2019). The flux of subtropical species is highest in the fall and winter, when stratification is high and upwelling relaxes (Figs. 4, 7, and 8). This is consistent with previous studies from the SBB sediment trap (Kincaid et al., 2000). The most abundant subtropical species (*G. rubescens*, *G. ruber*, *O. universa,* and *N. dutertrei*) are mixotrophs that rely on photosynthetic symbionts (Takagi et al., 2019). These species are adapted to succeed in oligotrophic conditions but are also found in more productive regions (Schiebel and Hemleben, 2017; Ward, 2019).
In the California Current system, the subtropical species are less abundant where light is limited close to shore due to high turbidity associated with primary productivity (Ortiz et al., 1995). We show that subtropical species, *G. rubescens, G. ruber*, *O. universa*, and *N. dutertrei*, are significantly correlated with SST in SBB (Fig. 5). Additionally, *G. ruber* is positively correlated with a deeper mixed layer, which is characteristic of the non-upwelling season when the water column is more stratified and the thermocline is gently sloping, extending deeper into the water column (Fig. 5). *Globigerinoides ruber* also
has a positive loading on CCA1, where upwelling related variables have negative loadings (Fig. 6 and Table 3).

We hypothesize that the combined extension of the upwelling season and slow warming of the California upwelling system has negatively impacted the abundance of subtropical species in SBB. As upwelling extends through the late summer or early fall, the optimal, warm SST season for subtropical species is shortened, increasing the amount of time where conditions are
unfavorable in SBB for subtropical species. This is supported by our observation of a shift in the seasonality for some subtropical species (Fig. 4). From 1993-1998, the average fluxes of *G. ruber* and *N. dutertrei* increase in May (Kincaid et al., 2000; Black et al., 2001; Fig. 4). This seasonal increase shifts to July in the samples from 2014-2021, which corresponds to an expansion of the upwelling season (Fig. 4).

**4.4 Impacts of El Niño**

There is no consistent response in the magnitude of foraminiferal flux in relation to El Niño events in SBB. The most consistent similarity across El Niño events is the increase in subtropical species flux. In SBB, El Niño events modify the structure of the water column by increasing SST and stratification (Lynn and Bograd, 2002). More southern-sourced waters enter the basin via the countercurrent and undercurrent, bringing warmer, saline, lower oxygen, nutrient-poor water (Bograd et al., 2001; Lynn





and Bograd, 2002; Fiedler and Talley, 2006; Auth et al., 2015). Surface productivity in SBB decreases during El Niño events,
but the response of foraminifera to these conditions is variable (Kincaid et al., 2000; Black et al., 2001; Catlett et al., 2021).
The periods 1993-1998 and 2014-2021 contain two strong and two weak El Niños (Trenberth et al., 2024). Foraminiferal flux
was exceptionally high during the 1996/1997 El Niño (Fig. 7) (Black et al., 2001). During the weak El Niño of 1993/1994,
foraminiferal flux was low (Kincaid et al., 2000). The 2015/2016 El Niño resulted in low foraminiferal flux, much like the
1993/1994 event (Kincaid et al., 2000; Figs. 2 and 7). However, the timeseries has a gap between July and October 2015,
limiting the interpretation of the 2015/2016 El Niño. During the weak El Niño in 2019, foraminiferal flux was not particularly
high or low (Figs. 2 and 7).

Subtropical species increased in abundance during El Niño events, consistent with the warming of the Eastern Tropical North
Pacific and advection of warmer waters into SBB via the enhanced inshore countercurrent and undercurrent (Kincaid et al.,
2000; Black et al., 2001; Lynn and Bograd, 2002; Fig. 2). The increase in stratification and SST during these years likely
provided favorable conditions for the subtropical species (Schiebel and Hemleben, 2017). The high spring foraminiferal flux
in May 1997 is interpreted by Black et al. (2001) as the result of a period of greatly reduced upwelling, where foraminifera
may have experienced less predation or competition for food. Interestingly, the heterotrophic *G. bulloides* followed similar
trends, with high flux during the 1997/1998 El Niño and increasing flux across the twentieth century in association with
warming and stratification (Black et al., 2001; Field et al., 2006; Takagi et al., 2019).

## 4.5 Upwelling indicator species

Certain species of foraminifera, including *G. bulloides*, *T. quinqueloba* and *G. glutinata*, are known to inhabit modern
upwelling environments and have been used as upwelling indicators in paleoceanographic reconstructions (Black et al., 2001;
Kucera, 2007; Sautter and Sancetta, 1992; Souto et al., 2011). Productivity is an important component of past environmental
conditions and has implications for marine carbon export and the entire marine food web (Capone and Hutchins, 2013). Using
foraminifera as upwelling indicators in the past can provide insight into changes in productivity (Sautter and Sancetta, 1992;
Kincaid et al., 2000; Kucera, 2007; Schiebel and Hemleben, 2017).

A good upwelling indicator should be most abundant during the upwelling season and have positive correlations with
upwelling-associated environmental parameters. An upwelling indicator that is applicable to the paleo record should respond
to long-term changes in upwelling in addition to seasonal variation. Historically, *G. bulloides* has been used as an upwelling
indicator, especially in the Arabian Sea where its abundance changes in accordance with the monsoon and changing wind
direction (Naidu and Malmgren, 1995; Gupta et al., 2003; Kucera, 2007). However, in the SBB record, *G. bulloides* flux is
neither positively correlated with upwelling season, nor upwelling-associated parameters (Figs. 2, 3, 4, and 5), and has
decreased while both the magnitude and duration of upwelling have increased (Jacox et al., 2018, Fig. 4).



A more promising upwelling indicator in SBB is *T. quinqueloba*. Consistent with Kincaid et al. (2000) and Black et al. (2001), *T. quinqueloba* responds more directly to upwelling than *G. bulloides* in our 2014-2021 timeseries (Figs. 2, 3, 4, 5). It is most abundant during the upwelling season, with peak abundance occurring later in 2014-2021 compared to 1993-1998, likely in accordance with the expansion of the upwelling season (Fig. 4). This species is significantly ($p < 0.05$) correlated with opal,
nitrogen, sample mass, organic carbon, pH, CUTI, and a shallow, steep thermocline that indicates the transport of cool water to the surface (Fig. 5). *Turborotalita quinqueloba* has a negative loading on CCA1, falling closest to organic carbon, opal, nitrogen, pH, and CUTI (Fig. 6). The median flux of *T. quinqueloba* has not changed significantly since the 1990s while most other species have decreased (Fig. 7). These changes drive the relative abundance increase in *T. quinqueloba* over the 28-year period, which is consistent with the increase in upwelling over the same period (Figs. 7 and 8).

**4.6 The value of long timeseries**

Comparing new data to historical baselines is a useful way to monitor the progression of climate change and its impact on marine ecosystems (Barry et al., 1995; Jonkers et al., 2019). The long-standing SBB sediment trap time-series collection since 1993 provides an opportunity to track changes in the foraminiferal population over time, as both global and local conditions vary. Our multi-year record also allows for a robust analysis of seasonal patterns beyond the scope of a single year, as it takes
interannual variability into account (Figs. 2, 3, and 4). For example, flux of *G. bulloides* appears seasonal in 1993-1998, but this is driven largely by a single El Niño event. A longer timeseries clarifies that *G. bulloides* is abundant year-round with variable timing in flux (Figs. 2 and 4). Thus, *G. bulloides* likely reflects annual average conditions in the sediment record. Species with distinct average seasonal patterns will be skewed in the sediment record towards the season during which they are most abundant (fall/winter for subtropical species and *N. incompta*, spring/summer for *T. quinqueloba*) (Figs. 4 and 9).
Long timeseries are thus needed in order to investigate the impacts of decadal variability and ongoing anthropogenic climate change on coastal ecosystems. This study focuses on rapid climate change over the past 28 years, but modes of decadal variability in the region (ENSO, PDO, NPGO) remain important for foraminifera and phytoplankton (Field et al., 2006; Osborne et al., 2020; Catlett et al., 2021). The combination of these modes of variability can act alongside anthropogenic climate change, intensifying or dampening various impacts (Osborne et al. 2020; Dalsin et al. 2023) and may require even
longer timeseries for their influence to be resolved.

**4.7 Implications for the paleontological record**

Due to their global distribution and regular contribution to the fossil record, foraminifera provide valuable information about climate change throughout geologic history (Lisiecki and Raymo, 2005; Kucera, 2007; Hönisch et al., 2012; Schiebel and Hemleben, 2017). As calcifiers, they are part of the marine carbon cycle and sensitive to ocean acidification (Schiebel, 2002;
Langer, 2008; De Moel et al., 2009; Moy et al., 2009; Davis et al., 2017; Osborne et al., 2016, 2020; Pallacks et al., 2023). This record of planktic foraminifera, bracketing 28 years of climate change within a productive, coastal region is ideal for



considering how rapid climate change events, such as those driven over the past century by anthropogenic $CO_2$ emissions, appear in the sediment record.

A decrease in foraminiferal flux (37.9%) and carbonate flux (21.9%) should be evident in the sediment record, given the sediment accumulation rate in SBB of 120 cm kyr$^{-1}$ (Behl, 1995). Depending on the sampling resolution, this would also bias climate records based on foraminiferal shells towards times when foraminifera were more abundant. Periods with lower foraminiferal flux due to ocean acidification, for example, would contribute fewer foraminifera to the sediment record, biasing a record that contains an acidification event towards the pre-acidification period.


In the geologic record, relative abundance is a common tool used to assess changes in the foraminiferal population over time, aiding interpretations of paleo-environmental and climatic conditions (Kennett and Ingram, 1995; Hendy and Kennett, 2000; Hendy et al., 2004; White et al., 2013). Between 1993-1998 and 2014-2021, the relative abundance of *G. bulloides* and subtropical species decreased, while relative abundance of *T. quinqueloba* and other non-subtropical species increased (Fig.

8). This is contrary to the warming trend found in SBB through the 20th century (Field et al., 2006), indicating a shift in the major drivers of foraminiferal abundance from temperature to ocean acidification between the 20th and 21st centuries. Based on assemblage data alone, the decrease in subtropical species abundance from 1993-2021 might be interpreted as cooling in the sediment record, which would be an incomplete picture given the influence of ocean acidification and expansion of the upwelling season in SBB in the context of global warming (Fig. 8). This study, which analyzes foraminiferal species flux data

alongside environmental conditions, highlights the value of including multiple proxies when interpreting records of warming or cooling. The interplay of circulation changes, ocean acidification, and warming continue (Gruber, 2011), but ocean acidification and the increase in regional upwelling seem to be the main drivers over the course of our study. As the California upwelling system is warming more slowly and acidifying faster than other marine regions (Feely et al., 2008; Hauri et al., 2009; Gruber et al., 2012; Seabra et al., 2019), the 28-year record may reflect a larger influence of ocean acidification compared

to temperature on recent decadal scales in SBB, with pH levels falling below the tolerance of many species.

**5 Conclusion**

Our 7-year sediment trap record of planktic foraminiferal flux in SBB from 2014-2021 provides an update to previously published records from 1993-1998 (Kincaid et al. 2000; Black et al. 2001). Over a period of 28 years, total foraminiferal flux and carbonate flux have dramatically decreased. This decrease in foraminiferal flux is likely driven by an increase in upwelling

and the acidification of the California coastal upwelling system. The decrease in carbonate export relative to organic carbon provides a negative feedback to atmospheric $CO_2$ and a strengthening of the biological pump. This timeseries has also allowed for multi-year analysis of seasonal patterns of major species, revealing *T. quinqueloba* as the most promising upwelling indicator in SBB. Additionally, the flux of subtropical species has decreased relative to 1993-1998. Although warming





continues, the intensification and expansion of upwelling in SBB and associated acidification have had a negative impact on

subtropical species, counter to what would be predicted from continued warming alone in SBB.









**Appendix A**

| Dataset | **Kincaid et al. (2000) and Black et al. (2001)** |
|---|---|
| Description | Foraminiferal records from the SBB sediment trap collected between 1993-1996 and 1996-1998, respectively |
| Data Obtained | Foraminiferal flux 1993-1998 |
| **Dataset** | **SBB Sediment Trap** |
| Description | Analysis of sediment trap material |
| Data | Fluxes of mass, organic carbon, carbonate, opal, terrigenous, nitrogen (g m$^{-2}$ d$^{-1}$) |
| Source | This paper |
| **Dataset** | **Coastal Upwelling Transport Index (CUTI)** |
| Description | CUTI estimates vertical water transport along the North American west coast. We used the values from the 1° bin at 34°N. |
| Data | Vertical water transport (m day$^{-1}$) |
| Source | Jacox, M. G., Edwards, C. A., Hazen, E. L, and Bograd, S. J. (2018). Coastal upwelling revisited: Ekman, Bakun, and improved upwelling indices for the U.S. west coast, *Journal of Geophysical Research*, 123(10), 7332-7350, doi:10.1029/2018JC014187. https://mjacox.com/upwelling-indices |
| **Dataset** | **NOAA National Data Buoy Center** |
| Description | Buoy data from SBB (46053 and 46054) |
| Data | Sea surface temperature (SST) |
| Source | NOAA National Data Buoy Center (1971). Meteorological and oceanographic data collected from the National Data Buoy Center Coastal-Marine Automated Network (C-MAN) and moored (weather) buoys [46053 and 46054]. NOAA National Centers for Environmental Information. Dataset. https://www.ncei.noaa.gov/archive/accession/NDBC-CMANWx. Accessed November 2023. |
| **Dataset** | **Plumes and Blooms** |
| Description | CTD and bottle data from station 4. |
| Data | CTD casts were used to calculate mixed layer depth, thermocline depth (greatest rate of temperature change), the depth of the bottom of the thermocline, and thermocline slope. Surface $PO_4$, $NO_3$ and $NO_2$ were gathered from bottle data. |
| Source | Southern California Bight MBON, D. Catlett, D. Siegel, and N. Guillocheau. 2022. Plumes and Blooms: Curated oceanographic and phytoplankton pigment observations ver 3. Environmental Data Initiative. https://doi.org/10.6073/pasta/29d4ef7976c19d6958e618d1548dcd72 (Accessed 2023-02-16). http://www.oceancolor.ucsb.edu/plumes_and_blooms |
| **Dataset** | **California Cooperative Oceanic Fisheries Investigations (CalCOFI)** |
| Description | CTD and bottle data from station 081.8, 046.9 |



| Data | CTD casts were used to calculate mixed layer depth, thermocline depth, the depth of the bottom of the thermocline, and thermocline slope. |
|---|---|
| Source | https://calcofi.org |
| **Dataset** | **Plumes and Blooms: phytoplankton pigment concentration** |
| Description | Phytoplankton pigment data from Plumes and Blooms cruises |
| Data | Chlorophyll *a*, Chlorophyll *b*, 19'-hexanoyloxyfucoxanthin (prymnesiophytes), fucoxanthin (diatoms), peridinin (dinoflagellates), and zeaxanthin (picophytoplankton) |
| Source | SCB Marine Biodiversity Observation Network, D. Catlett, D. Siegel, N. Guillocheau, and L. Kui. 2023. Plumes and Blooms: phytoplankton pigment concentration ver 3. Environmental Data Initiative. https://doi.org/10.6073/pasta/c6bbdf72bee131d00ad1bd24f5f74c87 (Accessed 2024-02-06). |
| **Dataset** | **Multistressor Observations of Coastal Hypoxia and Acidification (MOCHA)** |
| Description | Large, compiled dataset of oceanographic measurements along the US West Coast. Data was filtered by date (2014-2021) and coordinates within SBB. |
| Data | Salinity, pH, total alkalinity (TA), dissolved oxygen (DO), $PO_4$, $NO_3$ and $NO_2$ |
| Source | Kennedy, E. G., Zulian, M., Hamilton, S. L., Hill, T. M., Delgado, M., Fish, C. R., Gaylord, B., Kroeker, K. J., Palmer, H. M., Ricart, A. M., Sanford, E., Spalding, A. K., Ward, M., Carrasco, G., Elliott, M., Grisby, G. V., Harris, E., Jahncke, J., Rocheleau, C. N., Westerink, S., and Wilmot, M. I.: A high-resolution synthesis dataset for multistressor analyses along the US West Coast, Earth Syst. Sci. Data, 16, 219–243, https://doi.org/10.5194/essd-16-219-2024, 2024 |


**Table A1. List of additional datasets used and data obtained from each source.**





| Species | Original Description |
|---|---|
| *Globigerina bulloides* | d'Orbigny, 1826 |
| *Turborotalita quinqueloba* | Natland, 1938 |
| *Neogloboquadrina incompta* | Cifelli, 1961 |
| *Globigerinita glutinata* | Egger, 1893 |
| *Globorotalia scitula* | Brady, 1882 |
| *Globoturborotalita rubescens* | Hofker, 1956 |
| *Globigerinoides ruber* | d'Orbigny, 1839 |
| *Orbulina universa* | d'Orbigny, 1839 |
| *Neogloboquadrina dutertrei* | d'Orbigny, 1839 |
| *Globigerina falconensis* | Blow, 1959 |
| *Globigerinita uvula* | Ehrenberg, 1862 |
| *Neogloboquadrina pachyderma* | Ehrenberg, 1862 |
| *Globigerinoides tenellus* | Parker, 1958 |
| *Beella digitata* | Brady, 1879 |
| *Globigerinella calida* | Parker, 1962 |
| *Globigerinella siphonifera* | d'Orbigny, 1839 |
| *Bolivina variabilis* | Williamson, 1858 |
| *Tenuitellita parkerae* | Brönnimann & Resig, 1971 |
| *Globorotaloides hexagonus* | Natland, 1938 |
| *Tenuitellita iota* | Parker, 1962 |
| *Globoconella inflata* | d'Orbigny, 1839 |
| *Globorotalia truncatulinoides* | d'Orbigny, 1839 |
| *Trilobatus sacculifer* | Brady, 1877 |
| *Hastigerina pelagica* | d'Orbigny, 1839 |
| *Globorotalia hirsuta* | d'Orbigny, 1839 |




**Table A2: Full names and associated original description citations for all species found in SBB from 2014-2021.**






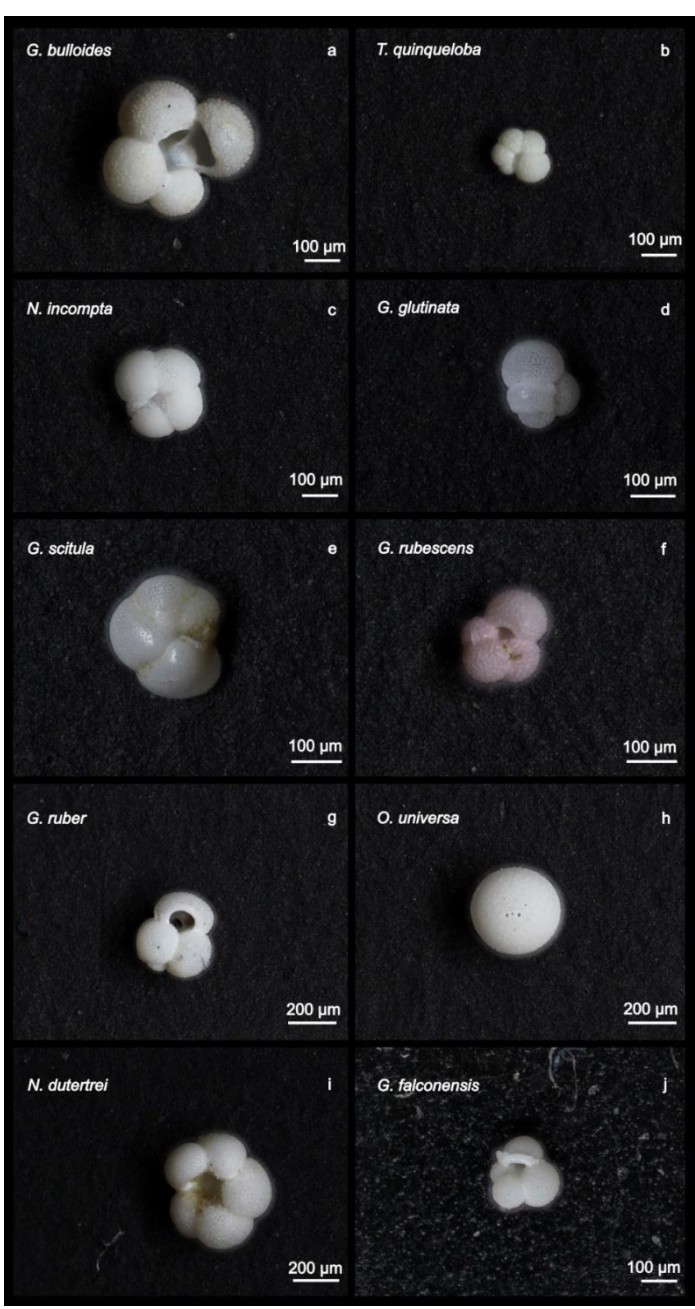

**Figure A1: Images of the 10 most abundant planktic foraminiferal species in SBB from 2014-2021.**




**Figure A2: Correlation matrix of rare species flux (x axis) with environmental parameters (y axis). Pearson correlation**
**coefficients (r) for statistically significant correlations (p < 0.05) are listed as numbers. Blue shades represent positive correlations, and red shades represent negative correlations. Gray represents correlation coefficients < 0.2. Presence of photosymbionts based on Takagi et al. (2019), Schiebel and Hemleben (2017), and Meilland et al. (2022) is represented by green circles.**







| Species | Rate of change (foraminiferal flux per year) May 1997 maximum removed | r |
|---|---|---|
| | | 695 |
| Total | -17.28 | 0.20 |
| *G. bulloides* | -14.18 | 0.28 |
| *T. quinqueloba* | 1.23 | 0.04 |
| *N. incompta* | -2.28 | 0.15 700 |
| *G. glutinata* | 0.83 | 0.32 |
| *G. scitula* | 0.55 | 0.25 |
| *G. rubescens* | -0.30 | 0.11 |
| *G. ruber* | -0.91 | 0.37 |
| *O. universa* | -0.58 | 0.33 |
| *N. dutertrei* | -2.32 | 0.52 705 |

**Table A3: Rates of change from 1993-2021 presented as foraminiferal flux per year (2nd column) and correlation coefficients (r) of the linear regression for each species (3rd column) with a single peak flux point from May 1997 removed.**




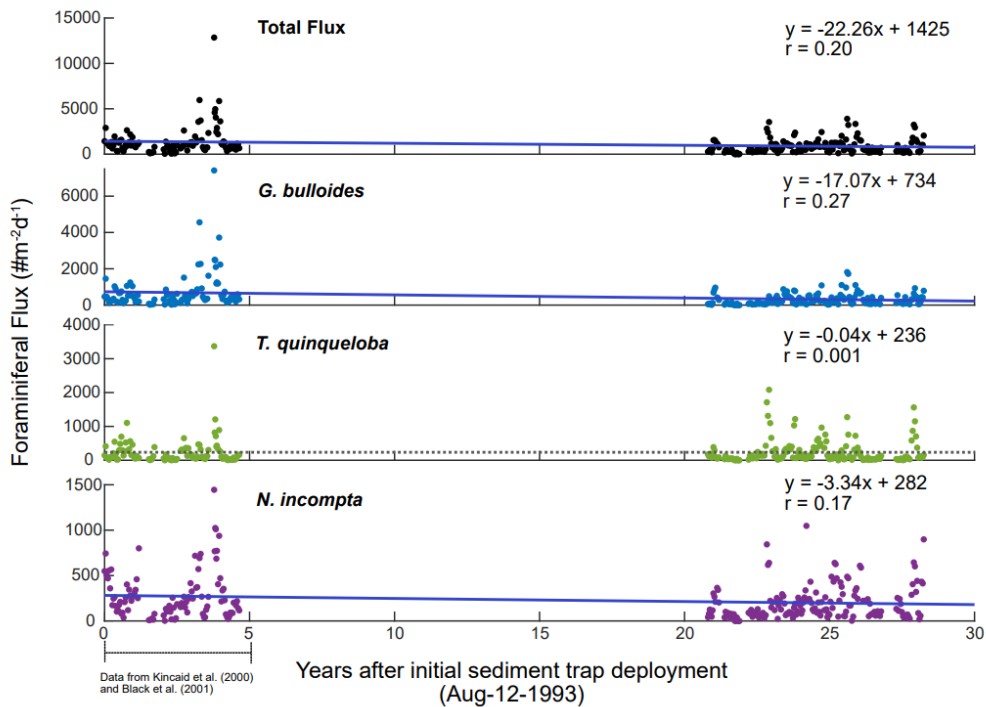

**Figure A3: Linear regression from 1993-2021 for total foraminiferal flux and the three most abundant species in SBB. Best fit lines represent rate of change of foraminiferal flux in units of foraminiferal flux per year.**





**Code availability**

Code used to generate figures is available upon request.

**Data availability**

All data is publicly available from https://www.bco-dmo.org/dataset/936276/

doi:10.26008/1912/bco-dmo.936276.1

**Sample availability**

Foraminiferal samples are archived at North Carolina State University. All other sediment trap material is archived at the
University of South Carolina.


**Author contributions**

C. Davis and C. Benitez-Nelson acquired funding and conceived the study. E. Havard and K. Cherry conducted data collection.
Data analysis and visualization was conducted by E. Havard. Resources and samples were provided by E. Tappa. E. Havard
wrote the original manuscript draft. C. Davis, C. Benitez-Nelson, E. Tappa, and K. Cherry reviewed and edited the manuscript.


**Competing interests**

The authors declare that they have no conflict of interest.

**Acknowledgements**

We would like to thank P. Fitts and S. Wall for their roles in sample processing, and R. Alcorn for providing foraminifera and
coccolithophore cartoons. We also want to thank E. Hyland and A. Schnetzer for their input and assistance. Thank you to the
captain and crew of the R/V Shearwater and everyone who has helped with sediment trap maintenance and sample collection.

**Financial support**

This research was funded by the National Science Foundation (OCE #2223074), to C Davis.



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
