# Peer review of "Decreasing foraminiferal flux in response to ongoing climate change in the Santa Barbara Basin, California"

_EGUsphere, 2024_

## Author Comment (AC3)

We would like to thank Reviewer 3 for their comments and suggestions that have helped us to clarify our manuscript.

Reviewer #3:

"I was invited as the third reviewer for the revised manuscript. The dataset and topic of this manuscript are both valuable and of broad interest to the paleoceanographic and plankton ecology communities. The authors have addressed the majority of the concerns raised by the initial reviewers in a generally sound manner.

However, the response to Reviewer 1's comments regarding the influence of environmental variables on foraminiferal fluxes, particularly for *G. bulloides*, remains insufficient (Lines 450-459). While the revised manuscript includes additional statistical explanations, it lacks a deeper exploration of the underlying ecological or biogeochemical mechanisms driving the observed patterns. In its current form, the discussion of *G. bulloides* flux is not only inconclusive but may also be confusing to readers. A more comprehensive interpretation of the potential drivers would substantially strengthen the manuscript and improve clarity. "

We have revised and elaborated through section 4.1 with an aim to further clarify potential drivers of shorter- and longer-term flux (Lines 431-480).

"In addition, while it is understandable that direct comparisons between the 2014–2021 and 1993–1998 periods may be complicated by temporal variability in environmental conditions, the authors should at least briefly acknowledge and discuss the potential implications of these long-term environmental changes for foraminiferal fluxes and assemblage composition."

We agree that it is important to acknowledge the implications of long-term changes in environmental conditions and foraminiferal flux. We have included additional information about the multi-decadal temperature and carbonate chemistry trends in discussion section 4.1, including a Santa Barbara harbor SST record (Lines 457-460) (Carter et al., 2022) and a carbonate chemistry record from CalCOFI station 90.90, located southwest of SBB (Lines 449-451) (Wolfe et al., 2023), which can hopefully also speak to the previous point raised. In section 4.2, we discuss a decrease in the ratio of inorganic to organic carbon flux and associated strengthening of the biological carbon pump as a key long-term implication of decreasing foraminiferal flux. We have added a summary line to section 4.2 to clarify and highlight this idea (Lines 499-501).

Minor suggestions:

"Please ensure that line thicknesses in all tables are consistent."

We have edited the line thickness of tables for consistency.

"Check all units throughout the manuscript and figures (e.g., there should be a space between "$m^{-2}$" and "$d^{-1}$")."

The spacing has been edited as suggested throughout the manuscript and figures.

"The subpanels in Figure 10 appear out of sequential order and should be rearranged accordingly."

The subpanels in Figure 10 have been labeled a, b, and c in sequential order.

References

Carter, M. L., Flick, R. E., Terrill, E., Beckhaus, E. C., Martin, K., Fey, C. L., Walker, P. W., Largier, J. L., McGowan, J.A.: Shore Stations Program, Santa Barbara (Santa Barbara Archive, 2025-03-14). In Shore Stations Program Data Archive: Current and Historical Coastal Ocean Temperature and Salinity Measurements from California Stations, https://doi.org/10.6075/J03N236M, 2022.

Wolfe, W. H., Martz, T. R., Dickson, A. G., Goericke, R., & Ohman, M. D. A 37-year record of ocean acidification in the Southern California current. *Communications Earth & Environment*, *4*(1), 406, https://doi.org/10.1038/s43247-023-01065-0, 2023.

---

## Author Response (AR1)

**Response to Associate Editor**

*Dear Emily Havard et al.,*

*Thank you for submitting your work to Biogeosciences. We have received two reports from external referees. Both commented on the potentially valuable contribution of your data set and work to the field, but they also raised critical points that require further attention and revisions. Key concerns include the relationships between G. bulloides and environmental parameters, comparisons between recent and older data sets, the effect of potential long-term changes in planktonic foraminifera size, and the discussion on the impact of ocean acidification, among others.*

*Based on the reviewer evaluations and your responses, I invite you to carefully consider the reviewers' comments and suggestions and make the necessary revisions. This is considered a major revision.*

*Additional notes the authors from review file validation:*
*"With the next file upload request, please number the tables and figures in the Appendix separately (Table A1 to Table A3 and Figure A1 to Figure A3)."*

*Best regards,*
*Yuan Shen*
*Associate Editor*

Dear Yuan Shen,

We thank you and the reviewers for your feedback that has allowed us to improve our manuscript. After consideration of the reviewers' evaluations, we have provided responses to each of their comments. Here, we provide a line-by-line description of edits made to our manuscript based on the suggestions of the reviewers. Additionally, we have updated the data table of sediment trap particle flux, including carbonate and organic carbon, to include data through October 2021 (previously January 2021), to exactly match the dates of our foraminiferal assemblage record. We have updated calculated values and Fig. 10 panel c as appropriate, and we have added the table to the supplement.

Sincerely,

Emily Havard

On behalf of all authors

**Edits in Response to Reviewer 1**

We have clarified the description of the relationships between *G. bulloides* and environmental parameters.

*Lines 452-456 "This species is further positively associated with surface dissolved oxygen, which has many possible drivers, including primary productivity, the properties of upwelled waters, and seasonal currents (Fig. 6). However, G. bulloides is negatively associated with environmental variables that are positively associated with upwelling (CUTI), including pH, organic carbon, nitrogen, and opal (Fig. 6). The negative association between G. bulloides and opal flux suggests that G. bulloides feeds on a variety of phytoplankton in SBB, rather than primarily on diatoms"*

We have clarified that acidification is a hypothesis presented as a possible explanation for the results observed in our study.

*Lines 467-469 "In the context of rapid acidification in the California Current system and an increase in local upwelling intensity, we hypothesize that the decrease in G. bulloides flux is most likely driven by ocean acidification such that conditions within SBB are moving beyond the range of tolerable conditions for this species."*

*Lines 622-623 "We hypothesize that this decrease in foraminiferal flux is driven by an increase in upwelling and the acidification of the California coastal upwelling system."*

We have added to the methods a description of long-term environmental data availability in SBB and long-term data used in our study.

*Lines 146-149 "Some environmental data from 1993-1998 in SBB was either non-existent or available at a temporal resolution too low to yield meaningful comparisons to the 2014-2021 environmental data. We thus discuss long term trends only in the available datasets of the Coastal Upwelling Transport Index (CUTI), foraminiferal flux, carbonate flux, and organic carbon flux."*

We have added a discussion of prey availability.

*Lines 456-459 "A change in prey availability could impact assemblage composition and foraminiferal flux, and more research is needed to determine the feeding dynamics of SBB species in detail. However, as G. bulloides is known to be an opportunistic feeder (Schiebel and Hemleben 2017), a shift in food type is unlikely to explain the extent of flux decrease observed in this study."*

We have included the sediment trap particle flux data table in the supplement.

**Edits in Response to Reviewer 2**

We have included a discussion of potential decrease in foraminiferal size as a driver of assemblage change.

*Lines 326-329 "Given the decrease in total flux, a potential decrease in foraminiferal size over time is important to consider. Between 1993-1998 and 2014-2021, the fluxes of the two smallest and most abundant species, T. quinqueloba and G. glutinata, remained consistent or increased slightly. Therefore, we conclude that a decrease in shell size to below the 125µm sieve size is not a major contributing factor for the observed decrease in total foraminiferal flux."*

We have clarified the post-mortem preservation of the tests in the methods section.

*Lines 126-129 "Sediment trap particles were preserved in a borate-buffered formalin solution (pH >8) in the sediment trap, minimizing interaction with the surrounding seawater. After trap recovery, samples were split, with a 1/16th split used for foraminiferal flux and species counts, excluding July-October 2015 and May-November 2020."*

We added information about benthic foraminifera and local sediment movement.

*Lines 188-189 "Benthic foraminifera contributed less than 0.5% of the total assemblage and there was no evidence of major resuspension events or landslides throughout the study period."*

We have added an explanation of the CCA loadings and directions to the methods section.

*Lines 160-162 "CCA loadings are the correlations between the canonical variables and environmental or species variables. For example, if a variable has a positive loading, it has a positive correlation with the canonical variable (CCA1 or CCA2) and is positively associated with other variables that have positive loadings on the same canonical variable."*

We have clarified to acknowledge the array of environmental factors that influence pteropod distribution and shell formation in addition to carbonate chemistry.

*Lines 66-71 "In addition to a variety of other environmental parameters such as salinity, oxygen, and temperature, pteropods are impacted by changes in carbonate chemistry (Bednaršek et al., 2019; Johnson et al., 2023; Meekes et al., 2021). Modern (2016) pteropods, for example,*

*produce thinner aragonite shells in the more acidic, nearshore upwelling zones of the California coast compared to offshore, due to a decrease in calcification (Mekkes et al., 2021). Foraminifera also calcify thinner shells in response to ocean acidification (De Moel et al., 2009; Moy et al., 2009; Osborne et al., 2016; Pallacks et al., 2023)."*

---

## Author Response (AR2)

**Response to Associate Editor**

*Dear Emily Havard et al.,*

*We have received an additional evaluation of your revised manuscript and responses. As the two original reviewers were unavailable, we invited a third reviewer to assess your submission. Overall, the reviewer recognized the potential value of your data and work, as well as the quality of the revised manuscript and your responses.*

*However, they also expressed concern regarding the insufficient explanation of the influence of environmental variables on foraminiferal fluxes, as well as the interannual comparison between the 2014–2021 and 1993–1998 periods.*

*I suggest that the authors carefully review the manuscript once more and make the necessary revisions to address these recurring concerns.*

*Best regards,*
*Yuan Shen*
*Associate Editor*

Dear Yuan Shen,

We thank you and Reviewer 3 for your feedback on our manuscript. Here, we provide a line-by-line description of edits made in response to the suggestions of Reviewer 3.

Sincerely,

Emily Havard

On behalf of all authors

**Edits in Response to Reviewer 3**

We have revised and elaborated throughout section 4.1 to further clarify the discussion of potential drivers of shorter-and longer-term flux. We have also included additional information about multi-decadal temperature and carbonate chemistry trends spanning the 1993-2021 time period.

*Lines 431-439, "The decrease in the median flux of G. bulloides (-58.8%) between 1993-1998 and 2014-2021 is the biggest contributor to the decline in total foraminiferal flux (-37.9%) (Fig. 7). Despite their decrease in numbers, G. bulloides has remained relatively abundant, inhabiting SBB year-round with a high degree of interannual variability, showing a lack of seasonal peaks when averaged over multiple years (Kincaid et al., 2000; Black et al., 2001; Figs. 2, 3, and 4). During 2014-2021, the species is positively associated with surface dissolved oxygen, which has many possible drivers, including primary productivity, the properties of upwelled waters, and seasonal currents (Fig. 6). Interestingly, G. bulloides is negatively associated with environmental variables that are positively associated with upwelling (CUTI), including pH, organic carbon, nitrogen, and opal flux (Fig. 6)."*

*Lines 441-443, "Few environmental datasets span the full 1993-2021 time period in SBB. However, available, nearby records of carbonate chemistry and temperature combined with particle and foraminiferal flux data from the SBB sediment trap allow us to make some reasonable inferences of possible drivers of the decrease in foraminiferal flux."*

*Lines 449-451, "At CalCOFI station 90.90, located on the western edge of the southern California Current, carbonate chemistry has been measured since 1983 (Wolfe et al., 2023). The measurements show a trend of decreasing pH by 0.0015 $yr^{-1}$ and decreasing carbonate ion concentration by 0.41 $\mu mol\ kg^{-1}\ yr^{-1}$ (Wolfe et al., 2023)."*

*Lines 457-460, "In contrast to changing carbonate chemistry, daily SST measurements taken in the Santa Barbara harbor show little change between 1993 and 2021. The mean SST from 1993-1998 was 16.3 ℃, while the mean SST from 2014-2021 was 16.8 ℃, with no apparent trend (Carter et al., 2022). Nor is warming evident at CalCOFI station 90.90 (Wolfe et al., 2023)."*

*Lines 467-477, "Environmental drivers such as acidification could also impact the broader ecosystem including prey species. However, G. bulloides is an opportunistic feeder (Schiebel and Hemleben, 2017), an inference which is further supported in SBB by its year-round presence despite differing availability of prey species (Catlett et al., 2021). Thus, while a change in prey*

*availability between the 1990s and 2020s cannot be entirely ruled out as a factor in assemblage composition, it is unlikely to be a primary driver of the decrease in foraminiferal flux. We hypothesize that both the negative association of G. bulloides with active upwelling indicators on short times scales as well as the long-term decrease in observed flux reflect the growing challenges of ocean acidification in this environment. Given the documented association between low pH and reduced calcification and stress in G. bulloides (Davis et al., 2017; Osborne et al., 2020), it may be that conditions within SBB are more frequently moving beyond the optimal range for this species."*

We have added a summary line to highlight a key implication of decreasing foraminiferal flux.

*Lines 499-501, "Although a decrease in calcification has negative consequences for calcifying organisms, a medium to long-term impact of a continued decrease in the observed carbonate flux may be an increase in the buffer capacity of the surface ocean, which may already be providing a stabilizing feedback to atmospheric $CO_2$ inputs."*